# Poverty and Influenza/Pneumococcus Vaccinations in Older People: Data from The Survey of Health, Ageing and Retirement in Europe (SHARE) Study

**DOI:** 10.3390/vaccines11091422

**Published:** 2023-08-27

**Authors:** Nicola Veronese, Nancy Zambon, Marianna Noale, Stefania Maggi

**Affiliations:** 1Geriatric Unit, Department of Internal Medicine and Geriatrics, Geriatrics Section, University of Palermo, Via del Vespro, 141, 90127 Palermo, Italy; 2Department of Economics and Management, University of Padova, 35122 Padova, Italy; nancy.zambon@unipd.it; 3Neuroscience Institute, Aging Branch, National Research Council, 35122, Padova, Italy; marianna.noale@in.cnr.it (M.N.); stefania.maggi@in.cnr.it (S.M.)

**Keywords:** vaccination, poverty, risk factors, SHARE

## Abstract

Vaccine acceptance seems to be lower in poor people. The determinants of the lower vaccine coverage in poor people are not established. Therefore, we aimed to explore the association between poverty and influenza/pneumococcus vaccinations and the factors potentially associated with vaccination’s coverage in poor people. The data of the Survey of Health, Ageing and Retirement in Europe (SHARE), an ongoing longitudinal, multi-disciplinary, and cross-national European study where used. Poverty was defined using information on income and household size. Among 47,370 participants initially included in the SHARE study, 12,442 were considered poor. In the multivariable logistic regression analysis, “Household size” was associated with a significantly lower vaccination probability, meanwhile “Age”, “Years of education”, “Regularly taking prescription drugs”, and the level of income were significantly associated with higher probabilities of both influenza and pneumonia vaccinations. The “Number of illnesses/health conditions” was significantly associated with a higher probability of getting vaccination against influenza and against pneumococcus. In conclusion, among poor older people, several specific factors could be identified as barriers for the vaccinations against influenza or pneumococcus that are unique to this segment of the population, such as living with the family and having a job.

## 1. Introduction

Influenza and pneumonia vaccinations are highly recommended in older people since they are effective not only in preventing these infections, but because they could also have some effects beyond the prevention of acute infections. For example, in the case of influenza vaccination, it has been widely reported that this vaccine is associated with a lower dementia risk [1] and a lower risk of cardiovascular diseases [2].

Vaccine acceptance, usually defined as the willingness to get vaccinated, is associated with numerous factors, e.g., the perceived risk of an infection, concerns regarding the efficacy of the vaccine, and its collateral effects. Recent studies regarding influenza vaccination have reported that, among all factors, the perceived risk associated with influenza and the belief in the efficacy of the vaccine are main factors affecting the acceptance of the vaccination, but the fear of adverse effects is the main deterrent [3]. These factors lead to a low coverage of vaccinations for influenza and for pneumococcus. Despite several policy recommendations, influenza vaccination uptake remains suboptimal. In Europe, for example, only one country reached the optimal cut-off of 75% among older people [4]. A similar picture was evident for the United States of America and in Oceania [5]. If we focus our attention on pneumococcus, as the data are even worse. One epidemiological study conducted in Italy pictured a scenario where, in the older adults, only 11% was covered by pneumococcal vaccinations, and 85% of older adults were not informed about vaccination opportunities and recommendations [6].

Since both influenza and pneumonia are easily preventable through vaccination, it is important to understand the prevalence of the various factors that may affect the willingness to get vaccinated in high-risk populations, such as the older adults. The literature suggests that in older people socio-economic status (SES) could be associated with a low vaccination coverage. A systematic review of SES and influenza vaccination in high-income countries reported very heterogenous results, with some studies showing a positive association between poverty and low influenza vaccination coverage, others without any significant association, and others reporting the contrary [7]. In this regard, the free access to vaccinations, guaranteed by the National Health Service in Italy and in other countries to older adults, has demonstrated that the socio-economic inequalities in influenza vaccine uptake were not present among the elderly citizens, while it was present in the rest of the population. Therefore, the different vaccination policies across countries could explain the heterogeneity on the association between SES and vaccines uptake reported in the literature [8]. At the same time, the ability of older adults to receive seasonal influenza vaccine could be influenced by social determinants, such as education, marital status, living alone, among many others [9,10]. However, clear epidemiological data regarding poverty and influenza and pneumococcus vaccinations are still scanty, particularly in a cross-national context. Finally, the factors associated with the acceptance and the access to vaccinations in poor people are not yet explored.

Given this background, we aimed to explore the association between poverty and influenza/pneumococcus vaccinations and the factors potentially associated with vaccination coverage in poor people using the data of the data of the Survey of Health, Ageing and Retirement in Europe (SHARE), an ongoing longitudinal, multi-disciplinary, and cross-national European study.

## 2. Materials and Methods

### 2.1. Study Details

We used data from SHARE, an ongoing longitudinal, multi-disciplinary, and cross-national European study. SHARE is a research infrastructure, funded by the European Union, for studying the effects of health, social, economic, and environmental policies over the life-course of European citizens [11]. During Waves 1 to 4, SHARE was reviewed and approved by the Ethics Committee of the University of Mannheim. Wave 4 and the continuation of the project were reviewed and approved by the Ethics Council of the Max Planck Society. In addition, the country implementations of SHARE were reviewed and approved by the respective ethics committees or institutional review boards whenever this was required. The numerous reviews covered all aspects of the SHARE study, including sub-projects, and confirmed the project to be compliant with the relevant legal norms and that the project and its procedures agree with international ethical standards (http://www.share-project.org/fileadmin/pdf_documentation/SHARE_ethics_approvals.pdf, accessed on 1 June 2023). Informed consent was obtained from all subjects involved in the study in written form.

The survey collects current and retrospective information on health, socio-economic status, and social and family networks of individuals aged fifty or older in (currently) twenty-seven European countries (plus Israel) [12]. SHARE started in 2004 and every second year a new regular survey has been run. Due to the COVID-19 outbreak, in March 2020 the regular (face-to-face) wave 8 data collection was suspended. Shortly after, two new additional telephone administered surveys were conducted in June–August 2020 (first SHARE-Corona survey, SCS1) and June–August 2021 (second SHARE-Corona survey, SCS2) with the aim to collect data on health and socio-economic impacts of COVID-19 among SHARE respondents [13].

### 2.2. Sample and Data

The current analyses used data from the two SHARE Corona Surveys. This allows us to account for different detailed characteristics, at the individual or household level, of individuals aged 50 or more (and their spouses or partners) living in Austria, Belgium, Bulgaria, Croatia, Cyprus, Czech Republic, Denmark, Estonia, Finland, France, Germany, Greece, Hungary, Italy, Latvia, Lithuania, Luxembourg, Malta, Netherlands, Poland, Portugal, Romania, Slovakia, Slovenia, Spain, Sweden, and Switzerland, plus Israel.

### 2.3. Focus on the Poverty

Several variables are available in the SHARE study for defining poverty. Using information on income and household size we computed square root scaled incomes, by dividing household income by the square root of household size, that in turn was used to classify respondents as Poor or Not Poor. Information regarding household income were collected using the following question asked only to the first household respondent in SCS1: “How much was the overall monthly income, after taxes and contributions, that your entire household had in a typical month before Corona broke out?”. For each country, we then labelled respondents belonging to the first quartile of the scaled income distribution as Poor, Not Poor, or otherwise.

### 2.4. Outcomes: Influenza and Pneumonia Vaccination

The outcomes of interest for this study are related to vaccinations commonly available in older persons and recommended by several national and international guidelines and societies. In SCS2, participants were asked to report, among other things, information regarding influenza and pneumonia vaccination. All respondents were asked the following questions: “In the last 12 months, did you get a flu vaccination?”, and “Did you have a pneumonia vaccination within the last six years, that is a pneumococcal vaccine?”. The possible answers were “yes”, and “no”.

### 2.5. Covariates

For assessing associations between individual and household (demographic, socio-economic, and health) characteristics and the outcomes of interest among the *Poor* SHARE respondents, we considered several factors. Our covariates included country of residence (categorized as mentioned before, the reference country for the country dummies is Germany); age (years, as continuous); gender; being in a couple (yes, no); years of formal education (as continuous); and household size (as continuous). From SCS1 we also recovered information about the job situation at the time when COVID-19 broke out (categorized as employed/self-employed vs. others); household’s ability to make ends meet since COVID-19 broke out (categorized as with great difficulty, with some difficulty, fairly easily, and easily); medication (regularly taking prescription drugs, categorized as yes, and no); self-perceived health before COVID-19 broke out (categorized as excellent, very good, good, fair, and poor); and household overall monthly income before COVID-19 broke out. Moreover, information regarding comorbidities was assessed in SCS2 by asking the participants: “Do you have any of the following illnesses or health conditions?”. Seven non-mutually exclusive options were presented to the respondents: hip fracture; diabetes or high blood sugar; high blood pressure or hypertension; heart attack including myocardial infarction or coronary thrombosis, or any other heart problem including congestive heart failure; chronic lung disease such as chronic bronchitis or emphysema; cancer or malignant tumour, including leukaemia or lymphoma, but excluding minor skin cancers; and any other illness or health condition. For the aim of this research, we compute the number of comorbidities reported by each respondent (continuous, ranging from 0 to 7).

### 2.6. Statistical Analysis

We first present descriptive statistics of our sample in terms of means and standard deviations (SD) of the key quantitative variables in use, while percentage and counts were considered for categorical variables.

Characteristics of the respondents were compared according to poverty and vaccination status using the two-sample *t* tests with unequal variances. Then, we ran multivariable logistic regressions with standard errors robust to the presence of heteroscedasticity to study the association between comorbidity and vaccination status among the respondents labelled as *Poor*. Thus, two different models were defined for both influenza and pneumonia vaccinations to assess whether (*Poor*) individuals affected by comorbidities have increased odds-ratios (ORs) for getting vaccinations. The first models, included an indicator for the “Number of illnesses/health conditions” (continuous variable ranging from 0 to 7), while the second model considered each of the seven illness/health conditions separately (binary indicators for: hip fracture; diabetes or high blood sugar; high blood pressure or hypertension; heart attack including myocardial infarction or coronary thrombosis or any other heart problem including congestive heart failure; chronic lung disease such as chronic bronchitis or emphysema; cancer or malignant tumour, including leukaemia or lymphoma, but excluding minor skin cancers; any other illness or health condition). All statistical tests were two-tailed, and a *p*-value < 0.05 was considered to be statistically significant. All analysis were conducted using STATA version 17.

## 3. Results

### 3.1. Sample Selection

We first selected respondents who were present in both SCS1 and SCS2. In total we had information on 48,342 respondents. We dropped 972 interviews with missing data on either influenza vaccination, pneumonia vaccination, self-perceived health, or comorbidities. This left a sample of 47,370 respondents (32,620 households).

### 3.2. Descriptive Statistics

Descriptive statistics for all variables in our final dataset of both *Not Poor*—34,928—and *Poor*—12,442—respondents are presented in Table 1. Comparing the sample by poverty, *Not Poor* respondents got influenza (39.6%) vaccination more often than *Poor* respondents (35.6% and 12.3%), respectively. These differences are statistically significant as indicated by the *p*-values (*t* tests with unequal variances, in the last column of Table 1) less than 0.0001. Respondents labelled as *Poor* were, with respect to their counterparts, more frequently females (65.3% vs. 56.1%), less often in a couple (52.8% vs. 76.5%), older (mean age: 72.0 vs. 69.4 years), less educated (mean years of formal education: 9.8 vs. 11.6), living in larger households (mean household size: 2.14 vs. 2.06), less often employed or self-employed (9.1% vs. 25.5%), had more difficulties in making ends meet, were more likely to be taking medications (79.95% vs. 74.99%), reported worse self-perceived health, and had lower monthly household income (average incomes: 992.47 € and 2119.53 € for *Poor* and *Not Poor*, respectively). Additionally, *Poor* respondents in our sample presented a higher incidence of comorbidities (average number of conditions 1.55, standard error 1.19) with respect to *Not Poor* respondents (average number of conditions 1.32, standard error 1.12). Moreover, *Poor* respondents also reported hip fracture more often (3.2% vs. 2.2%), diabetes or high blood sugar (20.0% vs. 16.1%), high blood pressure or hypertension (54.0% vs. 48.7%), heart attack (20.5% vs. 16.0%), chronic lung disease (8.6% vs. 6.1%), cancer or malignant tumour (5.9% vs. 5.5%), and any other illness or health condition (42.7% vs. 37.9%). All these comparisons were statistically significant at a *p*-value < 0.05.

Appendix A present descriptive statistics for the subsample of *Poor* respondents by vaccination status, while Appendix A reports results for the full subsample (12,442 respondents), and Appendix A shows results for the same subsample but excluding Romania and Slovakia since in these two countries no respondent reported pneumonia vaccination (11,860 respondents). The results in Table 2 and Table 3 show that people having received the vaccinations were significantly more frequently in a couple, older, in smaller households, less frequently employed or self-employed, reported (on average) less difficulties in making ends meet, more likely to be taking medications, and had a higher household income than their counterparts. Moreover, participants with either influenza or pneumonia vaccination reported a significantly higher prevalence of comorbidities, both in terms of average number and as percentage of each illness/condition. All these comparisons were statistically significant at a *p*-value < 0.001.

### 3.3. Logistic Regression Outcomes

We report our findings in Table 2 and Table 3 by showing the OR and 95% Confidence Interval (CI) for the logistic regression models we estimated; models reported in Table 2 include an indicator for the “Number of illnesses/health conditions” (continuous variable ranging from 0 to 7), while regressions reported in Table 3 consider each of the seven illness/health condition separately (binary indicators for: hip fracture; diabetes or high blood sugar; high blood pressure or hypertension; heart attack including myocardial infarction or coronary thrombosis or any other heart problem including congestive heart failure; chronic lung disease such as chronic bronchitis or emphysema; cancer or malignant tumour, including leukaemia or lymphoma, but excluding minor skin cancers; any other illness or health condition). While “Household size” was associated with a significantly lower vaccination probability, “Age”, “Years of education”, “Regularly taking prescription drugs”, and the level of income (“Log (income before COVID-19)”) were significantly associated with higher probabilities of both influenza and pneumonia vaccinations. Interestingly, while “Make Ends Meet” with some difficulty or fairly easily were significantly associated with influenza vaccination, they were not associated with pneumonia vaccination. Moreover, the “Number of illnesses/health conditions” was significantly associated with a higher probability of getting vaccination against influenza (Table 2 OR = 1.187, *p* < 0.001) and a higher probability of getting vaccination against pneumococcus (Table 2 OR = 1.213, *p* < 0.001).

## 4. Discussion

In this study including about 50,000 European participants to the SHARE project, we found that the prevalence of influenza and pneumonia vaccinations among poor people was significantly lower compared to not poor. Moreover, among poor people, we identified several potential barriers for vaccinations that could be possible targets of public health interventions.

In the SHARE study, the overall prevalence of influenza and pneumococcus vaccinations was 38.5% and 12.9%, respectively, indicating that the actual coverage is still sub-optimal in several European countries. Moreover, comparing the sample by the presence of poverty, not-poor respondents received the influenza and pneumococcus vaccinations more often than poor respondents, indicating overall that, in Europe, poverty could be a risk factor for lower vaccinations’ rates.

Poor participants receiving the vaccinations were older, less frequently employed or self-employed, more likely to be taking medications, and more likely to have comorbidities. Some of these factors were already explored by the literature. For example, comorbidities and older age are important factors that increase the likelihood of vaccination, probably reflecting the fact that frailer older individuals may have more frequent contacts with healthcare providers and an increased likelihood of being offered the vaccinations [14]. Moreover, higher educational level was positively associated with vaccinations, as expected, since people having a higher educational level have also a higher health literacy, and therefore can better understand the role of vaccines in disease prevention [15]. At the same time, our study suggests that poor older people living with the family had a significantly lower probability of being vaccinated for both pneumococcus and influenza, therefore increasing the risk of diseases’ transmission to other potential high-risk groups, such as children. Interestingly, living alone has been reported to be an important barrier for vaccinations [10,14]; however, when focusing on the poor segment of the population, where the household size is inversely associated with the income, living with the family decreases the chances of being vaccinated. Finally, our work shows that having a job is also negatively associated with vaccination, probably because it implies a better health status, and therefore fewer chances to have contacts with health care professionals and receiving adequate recommendations.

Whilst low SES and poverty have been widely reported to be associated with a lower vaccination rate [16,17,18], the factors that lead to vaccine uptake among the poor segments of the population are largely unknown. Considering that continuous efforts should be made to vaccinate everyone who is eligible, it remains a priority to vaccinate older adults. However, they represent a very heterogeneous group in terms of SES and this fact needs to be taken into account when designing strategies to promote vaccine uptake. Therefore, we believe that our work may be relevant because it identified factors associated with vaccination coverage in the poor segment of older adults, a particularly socially vulnerable and hard to reach subgroup.

A recent systematic review including 24 studies reported that in older people effective interventions are awareness campaigns, incentives, or easier access to vaccination, and that combining interventions is probably more effective that single interventions [19]. It should be noted that all these interventions target factors that our work has identified as barriers in the poor older adults.

Finally, our work also suggests that some factors are potentially different between pneumonia and flu vaccinations. In particular, “Make Ends Meet” with some difficulty or fairly easily were significantly associated with influenza vaccination, but not with pneumonia vaccination. We can justify this finding suggesting that household decisions have an important role in seasonal vaccination, such as flu, as compared to pneumonia.

In conclusion, we believe that our study may have several novel findings compared to the previous literature. In particular, to the best of our knowledge, it is the first investigation to specifically address the possible risk factors for lower vaccinations’ coverage in the poor segment of the older population, instead of exploring poverty as potential risk factor for low coverage.

Our findings must be interpreted within some limitations. First, several pieces of information were self-reported, such as those regarding comorbidities and medications. Second, influenza and pneumococcus vaccinations were recorded only once (i.e., past 12 months) during the SHARE study: therefore, it is also possible that older people not undergoing vaccination at this time received vaccinations during the previous years. This would be particularly relevant for pneumococcal vaccination, that, to the contrary of influenza vaccination, is not recommended on a yearly basis. Finally, the data collected refer to a specific time point during the COVID-19 pandemic that could have affected our findings. On the contrary, the major strengths of this study are undoubtedly the large cohort and the large number of countries representative of Europe and Israel.

## 5. Conclusions

Our work showed that, among poor older adults, several specific factors could be identified as barriers for the vaccinations against influenza or pneumococcus that are unique to this segment of the population, such as living in the family and being healthy and still working. Our findings indicate the necessity of intervention studies that could address these factors and verify whether they are effective in increasing vaccination coverage in this high-risk segment of the population. Although we focused only on influenza and pneumococcus pneumonia, our results could be extended to all vaccinations recommended in the adult population, such as for herpes zoster, pertussis, and COVID-19, for which the data on the coverage rates are scanty in all countries included in the study in spite of the heavy associated-burden of disease. In conclusion, we believe that the potential public health impact of these results goes well beyond influenza and pneumonia.

## Figures and Tables

**Table 1 vaccines-11-01422-t001:** Descriptive statistics by poverty.

Variable	*Not Poor* (n = 34,928)		*Poor* (n = 12,442)		*p*-Value
Influenza vaccination (%)	39.57		35.56		<0.0001
Pneumonia vaccination (%)	13.14		12.27		0.0116
Gender (female) (%)	56.11		65.25		<0.0001
Couple (%)	76.49		52.83		<0.0001
Mean age, years (SD)	69.39	(8.87)	71.97	(9.33)	<0.0001
Mean years of education (SD)	11.64	(4.19)	9.76	(3.92)	<0.0001
Mean household size (SD)	2.06	(0.84)	2.14	(1.23)	<0.0001
Employed/self-employed (%)	25.48		9.13		<0.0001
Make ends Meet					
1 - With great difficulty (%)	5.49		19.76		
2 - With some difficulty (%)	23.35		33.12		
3 - Fairly easily (%)	36.81		31.68		
4 - Easily (%)	34.35		15.44		
Regularly taking prescription drugs (%)	74.99		79.95		<0.0001
Self-perceived health					
1 - Excellent (%)	7.68		4.46		
2 - Very good (%)	18.46		12.07		
3 - Good (%)	45.33		43.52		
4 - Fair (%)	23.47		31.22		
5 - Poor (%)	5.06		8.72		
Mean household monthly income €	2119.53	(1631.80)	992.47	(744.61)	<0.0001
Mean number of illnesses and health conditions	1.32	(1.12)	1.55	(1.19)	<0.0001
Hip fracture (%)	2.19		3.21		
Diabetes/high blood sugar (%)	16.06		20.00		
High blood pressure/hyp. (%)	48.65		54.04		
Heart attack (%)	16.02		20.51		
Chronic lung disease (%)	6.07		8.59		
Cancer or malignant tumour (%)	5.46		5.92		
Other illness/health condition (%)	37.89		42.65		

**Table 2 vaccines-11-01422-t002:** Factors significantly associated with flu and pneumonia vaccination in poor people—estimated odds-ratios (with 95% CI).

Variables	Outcome: Flu Vaccination	Outcome: Pneumonia Vaccination
Female	1.003	1.091
	(0.914–1.101)	(0.956–1.245)
Couple	1.339 ***	1.070
	(1.195–1.501)	(0.904–1.266)
Age (years)	1.234 ***	1.432 ***
	(1.148–1.326)	(1.281–1.602)
Age × Age	0.999 ***	0.998 ***
	(0.998–0.999)	(0.997–0.998)
Years of education	1.011 *	1.021 **
	(0.999–1.024)	(1.004–1.039)
Household size	0.944 **	0.912 **
	(0.899–0.992)	(0.837–0.993)
Employed/self-employed	0.743 ***	0.878
	(0.620–0.890)	(0.679–1.136)
With some difficulty	1.205 ***	1.089
	(1.054–1.377)	(0.873–1.357)
Fairly easily	1.205 **	1.164
	(1.036–1.403)	(0.909–1.490)
Easily	1.445 ***	1.613 ***
	(1.207–1.731)	(1.231–2.113)
Regularly taking prescription drugs	2.021 ***	1.727 ***
	(1.778–2.298)	(1.410–2.115)
Self-perceived health		
Very good	1.069	1.422 **
	(0.843–1.356)	(1.029–1.966)
Good	1.255 **	1.309 *
	(1.009–1.562)	(0.964–1.777)
Fair	1.153	1.319 *
	(0.916–1.450)	(0.956–1.819)
Poor	1.217	1.660 ***
	(0.934–1.585)	(1.145–2.408)
Log (income before COVID-19)	1.195 *	1.515 **
	(0.981–1.457)	(1.102–2.082)
Number of illnesses/health conditions	1.187 ***	1.213 ***
	(1.138–1.237)	(1.144–1.286)
Observations	12,442	11,860

Regressions include an indicator for the “Number of illnesses/health conditions” (continuous variable ranging from 0 to 7). Robust CI in parentheses. *** *p* < 0.01, ** *p* < 0.05, * *p* < 0.1.

**Table 3 vaccines-11-01422-t003:** Factors significantly associated with flu and pneumonia vaccination in poor people—estimated odds-ratios (with 95% CI)-using single medical conditions.

Variables	Outcome: Flu Vaccination	Outcome: Pneumonia Vaccination
Female	1.011	1.128 *
	(0.921–1.111)	(0.987–1.290)
Couple	1.346 ***	1.082
	(1.201–1.509)	(0.913–1.281)
Age (years)	1.223 ***	1.443 ***
	(1.138–1.315)	(1.286–1.619)
Age × Age	0.999 ***	0.998 ***
	(0.998–0.999)	(0.997–0.998)
Years of education	1.012 *	1.022 **
	(0.999–1.024)	(1.005–1.041)
Household size	0.942 **	0.904 **
	(0.897–0.990)	(0.829–0.986)
Employed/self-employed	0.743 ***	0.904
	(0.620–0.890)	(0.699–1.169)
Make ends Meet		
With some difficulty	1.204 ***	1.085
	(1.053–1.376)	(0.870–1.353)
Fairly easily	1.209 **	1.185
	(1.038–1.407)	(0.924–1.521)
Easily	1.446 ***	1.668 ***
	(1.207–1.732)	(1.271–2.190)
Regularly taking prescription drugs	1.990 ***	1.827 ***
	(1.747–2.266)	(1.489–2.243)
Self-perceived health		
Very good	1.074	1.444 **
	(0.847–1.362)	(1.046–1.993)
Good	1.263 **	1.331 *
	(1.015–1.572)	(0.982–1.805)
Fair	1.168	1.324 *
	(0.928–1.470)	(0.960–1.826)
Poor	1.238	1.584 **
	(0.949–1.615)	(1.087–2.307)
Log (income before COVID-19)	1.197*	1.558 ***
	(0.982–1.459)	(1.131–2.146)
Hip Fracture	0.907	1.272
	(0.718–1.146)	(0.933–1.735)
Diabetes/high blood sugar	1.248 ***	1.144 *
	(1.121–1.389)	(0.984–1.331)
High blood pressure/hypertension	1.257 ***	1.095
	(1.144–1.380)	(0.960–1.249)
Heart attack	1.108 *	1.184 **
	(0.993–1.236)	(1.016–1.381)
Chronic lung disease	1.488 ***	2.827 ***
	(1.279–1.732)	(2.337–3.420)
Cancer or malignant tumour	1.143	1.297 **
	(0.955–1.368)	(1.032–1.630)
Other illness/health condition	1.080 *	0.917
	(0.988–1.181)	(0.807–1.043)
Observations	12,442	11,860

Regressions consider each of the seven illnesses/health conditions separately. Robust CI in parentheses. *** *p* < 0.01, ** *p* < 0.05, * *p* < 0.1.

## Data Availability

The database is available upon reasonable request to the corresponding author.

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
