# Peer review of "Poverty and Influenza/Pneumococcus Vaccinations in Older People: Data from The Survey of Health, Ageing and Retirement in Europe (SHARE) Study"

_vaccines, 2023, doi:10.3390/vaccines11091422_

Round 1

Reviewer 1 Report

This a very interesting and well written study on factors associated with the acceptance of vaccines in older people based on the income of the household.

The study, based on a quite large and robust database suggests a series of well defined variables able to predict the acceptance of flu vaccination and pneumococcal vaccination. 

In fact, I've no substantial requests for this paper. Only two minor suggestions.

- please include "years" when reporting age 

- Table 2 is informative but the captions are still misleading. I would suggest to break it down in two distintive table (including separately column 1+3 and 2+4) in order to simplify the conveyed message.

Thank you for your efforts with this interesting study

Author Response

This a very interesting and well written study on factors associated with the acceptance of vaccines in older people based on the income of the household. The study, based on a quite large and robust database suggests a series of well defined variables able to predict the acceptance of flu vaccination and pneumococcal vaccination. In fact, I've no substantial requests for this paper.

We thank the reviewer for the appreciation on our work.

Only two minor suggestions.

- please include "years" when reporting age

As suggested, “years” has been now added when reporting age.

- Table 2 is informative but the captions are still misleading. I would suggest to break it down in two distintive table (including separately column 1+3 and 2+4) in order to simplify the conveyed message.

We thank the Reviewer for the suggestion; we have now divided Table 2 into two distinct tables (Tables 2 and Tables 3).

Thank you for your efforts with this interesting study.

Reviewer 2 Report

This paper is reasonably well written and the analyses it reports appear to be well done. Here are a couple of items to attend to in a revision.

First, in Section 2.5 Statistical Analysis, in is stated: "Then, we run multivaria- ble logistic regressions with standard errors robust to the presence of heteroscedasticity to study the association between comorbidity and vaccination status among the respondents labelled as Poor. It would be good to have more detail about the specifications of the logistic regressions. You could add a sentence here that indicates that the specific content of the regressions is described below in the text related to the contents of the Table 2. 

Second, in the text description of the regression results reported in Table 2, you need to note that all levels of the Make Ends Meet set of regressors have statistically significant effect coefficients for the Flu Vaccination outcome variable but not for the Pnuemonia Vaccination outcome (not significant for the with some difficulty category). The may be indicative of household decisions on the seasonal likelihood of Flu contagion as compared to pnuemonia risk.

Third, in Section 5, it would be good to state the unique contributions of the research reported in the sense of how it goes beyond and extends the findings of prior studies. 

The English text needs only minor editing. 

Author Response

This paper is reasonably well written and the analyses it reports appear to be well done. Here are a couple of items to attend to in a revision.

First, in Section 2.5 Statistical Analysis, in is stated: "Then, we run multivaria- ble logistic regressions with standard errors robust to the presence of heteroscedasticity to study the association between comorbidity and vaccination status among the respondents labelled as Poor. It would be good to have more detail about the specifications of the logistic regressions. You could add a sentence here that indicates that the specific content of the regressions is described below in the text related to the contents of the Table 2.

We have now added a sentence describing logistic models developed more in details (pages 3-4), as follows:

“…two different models were defined for both influenza and pneumonia vaccinations to assess whether (Poor) individuals affected by comorbidities have increased odds-ratios (OR) for getting vaccinations: the first models, presented in Table 2, included an indicator for the “Number of illnesses/health conditions” (continuous variable ranging from 0 to 7), while the second models, presented in Table 3, considered each of the seven illness/health condition separately (binary indicators for: hip fracture; diabetes or high blood sugar; high blood pressure or hypertension; heart attack including myocardial infarction or coronary thrombosis or any other heart problem including congestive heart failure; chronic lung disease such as chronic bronchitis or emphysema; cancer or malignant tumour, including leukaemia or lymphoma, but excluding minor skin cancers; any other illness or health condition)”

Second, in the text description of the regression results reported in Table 2, you need to note that all levels of the Make Ends Meet set of regressors have statistically significant effect coefficients for the Flu Vaccination outcome variable but not for the Pneumonia Vaccination outcome (not significant for the with some difficulty category). The may be indicative of household decisions on the seasonal likelihood of Flu contagion as compared to pneumonia risk.

We thank the Reviewer for the possibility to clarify this point. We have specified in the Results paragraph this aspect (page 6).

“Interestingly, while “Make Ends Meet” with some difficulty or fairly easily were significantly associated with influenza vaccination, they were not associated with pneumonia vaccination.”

Finally, we added the following explanation in the Discussion section  (page 9):  

“Finally, our work also suggests that some factors are potentially different between pneumonia and flu vaccinations. In particular, “Make Ends Meet” with some difficulty or fairly easily were significantly associated with influenza vaccination, but not with pneumonia vaccination. We can justify this finding suggesting that household decisions have an important role for seasonal vaccination, such as flu, as compared to pneumonia.”

Third, in Section 5, it would be good to state the unique contributions of the research reported in the sense of how it goes beyond and extends the findings of prior studies.

We sincerely thank the Reviewer for giving us the possibility to further expand our findings. We have added this sentence in the Discussion section (page 9):

“In this regard, we believe that our study may have several novel findings compared to the previous literature. In particular, to the best of our knowledge, it is the first investigation to specifically address the important topic of the possible risk factors for lower vaccinations’ coverage in poverty, instead of exploring poverty as potential risk factor for low coverage. Our findings might be important for interventions better tailored for poor people for increasing vaccination coverage.”    

Reviewer 3 Report

- Line 55-57: this sentence is ambiguous. How was free access a major determinant for "these differences"? I tried my best to understand what you wanted to say, but I failed.

- The Introduction focused on psychosocial predictors/correlates of vaccine acceptance/uptake among older adults. It did not refer to the socio-demographic predictors at all. Please remember that SHARE data has no psychosocial variables.

- Line 82-85: the countries should be arranged alphabetically.

- Line 106: Why was Germany used as a reference country? Why not the avg EU-27?

- Line 145: Does the statistically significant difference between (13.1% vs 12.3%) and (39.6% vs 35.6%) mean anything clinically or practically?!
The same applies to all data in Table 1. Given the massive sample size of SHARE data, those binomial analyses are super misleading if considerable clinical differences do not support them.

- Was Table 2 based on the entire sample outcomes or only Poor individuals?If it is based on the entire sample, then this analysis is unrelated to the study context and objective. If it is based on Poor individuals only, then it can not answer the genuine question of the study because you need to compare the  Poor vs Not Poor to highlight the significant predictors of vaccination among lower SES groups as compared to higher SES groups.

The entire manuscript should undergo a native language check.

Author Response

- Line 55-57: this sentence is ambiguous. How was free access a major determinant for "these differences"? I tried my best to understand what you wanted to say, but I failed.

We are extremely sorry for the inconvenience. We have better explained, as follows, the previous sentence (page 2):

“In this regard, the free access to vaccinations, guaranteed by the National Health Service in Italy to older adults, has demonstrated that the socio-economic inequalities in influenza vaccine uptake were not present among the elderly citizens, while it was present in the rest of the population. Therefore, the different vaccination policies across countries could explain the heterogeneity on the association between SES and vaccines uptake reported in the literature. [8]”

- The Introduction focused on psychosocial predictors/correlates of vaccine acceptance/uptake among older adults. It did not refer to the socio-demographic predictors at all. Please remember that SHARE data has no psychosocial variables.

Thank you for your comment. We have now revised the Introduction in order to make more adherent to the aims of the work.

- Line 82-85: the countries should be arranged alphabetically.

Countries are now alphabetically listed, as suggested.

- Line 106: Why was Germany used as a reference country? Why not the avg EU-27?

Thank you for the question. Germany was arbitrarily used as reference country since this country in the SHARE context is a good representation of European people in terms of demographic factors.

- Line 145: Does the statistically significant difference between (13.1% vs 12.3%) and (39.6% vs 35.6%) mean anything clinically or practically?! The same applies to all data in Table 1. Given the massive sample size of SHARE data, those binomial analyses are super misleading if considerable clinical differences do not support them.

We fully agree with this request. In this sense, we would like to observe that we have now reported only the descriptive results statistically significant at a p-value <0.0001, instead of <0.05 for better accounting to the large sample size. Moreover, in our opinion, it is hard to know when a difference between two groups is clinically significant, in terms of prevalence.     

- Was Table 2 based on the entire sample outcomes or only Poor individuals? If it is based on the entire sample, then this analysis is unrelated to the study context and objective. If it is based on Poor individuals only, then it can not answer the genuine question of the study because you need to compare the  Poor vs Not Poor to highlight the significant predictors of vaccination among lower SES groups as compared to higher SES groups.

We are sorry for this inconvenience. In current Table 2 and 3, we have reported in the titles that the analyses regard only poor people and not the entire cohort. As reported at the end of the Introduction (page 2 and 3), with our analyses we aimed to determine characteristics associated with vaccinations among poor individuals, to identify possible targets for public health interventions to increase vaccination in this segment of the population.

Round 2

Reviewer 2 Report

The revisions to this manuscript have been responsive to the previous review. I have no further suggestions for revision. 

The English language generally is acceptable and requires only a minor polishing edit.

Author Response

Thank you for your appreciation: we have modified our paper, in order to meet the word count. 

Reviewer 3 Report

Thank you for addressing each of my earlier comments properly and satisfactorily. The manuscript has been improved significantly.

Thank you for addressing each of my earlier comments properly and satisfactorily. The manuscript has been improved significantly.

Author Response

(The authors gave the same response as above.)
